# The Effects of Receiving and Expressing Health Information on Social Media during the COVID-19 Infodemic: An Online Survey among Malaysians

**DOI:** 10.3390/ijerph19137991

**Published:** 2022-06-29

**Authors:** Hongjie (Thomas) Zhang, Jen Sern Tham, Moniza Waheed

**Affiliations:** Department of Communication, Universiti Putra Malaysia, Serdang 43400, Selangor, Malaysia; gs63096@student.upm.edu.my (H.Z.); moniza@upm.edu.my (M.W.)

**Keywords:** health information overload, Cognitive Mediation Model, information engagement, information processing, COVID-19

## Abstract

Social media is used by the masses not only to seek health information but also to express feelings on an array of issues, including those related to health. However, there has been little investigation on the influence of expressing and receiving information in terms of health-related engagement on social media. Moreover, little is known of the cognitive mechanisms underlying the expression and reception of health information on information overload (IO) during an infectious disease outbreak. Guided by the Cognitive Mediation Model (CMM), this study proposes a conceptual model to understand the effects of receiving and expressing COVID-19 information on social media on IO. Using an online survey conducted in Malaysia, our results indicated that risk perception positively predicted the reception and expression of information which, in turn, was positively associated with perceived stress and IO. Additionally, perceived stress triggered IO, indicating that the greater the perceived stress from health information engagement, the higher the likelihood of one experiencing IO during the pandemic. We conclude that the CMM can be extended to study IO as an outcome variable. More studies in diverse health contexts need to be conducted to enhance the conceptualization and operationalization of IO in health information processing.

## 1. Introduction

The COVID-19 pandemic has affected all areas of our lives in the past two years. Many countries had to lockdown or implemented drastic control measures to reverse the COVID-19 pandemic in the communities and keep it at bay. When the public situates in the era of the COVID-19 pandemic, social media platforms have become the dominant avenue for most of them to access different types of COVID-19 information and remain connected with their social networks [1,2,3,4,5]. Individuals rely on various social media platforms to access daily COVID-19 case updates, understand vaccination-related knowledge and health policies, follow the latest preventive measures announced by the authorities, and communicate with family members or friends who are unable to meet physically due to restrictions on traveling and gathering. Apart from the widely acknowledged popularity and contributions of social media, the development of social media can create new problems. For example, social media made it challenging for public health authorities to manage the pandemic and design health messaging to promote preventive strategies among the public [3]. At this time, various types of COVID-19 mis/dis/mal-information are freely spreading on various social media platforms without censorship, including various conspiracy theories about the virus and vaccination, misleading information about how to treat COVID-19 infections, and scientifically incorrect messages on preventive measures. As the social media platforms offer a mixture of endorsed health information and conflicting information to the public every day, the World Health Organization (WHO) has indicated that the COVID-19 pandemic is accompanied by a true social media infodemic [6]. 

During this trying time, truth and rumors about the virus continue to spread on social media, causing its users to feel uncertain, confused, fatigued, and overloaded with COVID-19 health information [4]. It is not an overstatement that social media has opened up for rich health information acquisition, unprecedented in the history of the public health crisis that combines technology and social media to keep the public informed [7]. Contrary to conventional media in which the information tends to be selected by the gatekeepers and serves the local community [8], during the infodemic, social media presents myriad types of health information about COVID-19 to anyone and anywhere. This is detrimental to individuals’ health outcomes if they engage in ineffective and incorrect remedies or recommendations. In response, social media users operate within a complex information environment, featuring a wealth of ill-founded information that can overwhelm them and leave them experiencing different negative feelings [9]. These feelings reduce the effectiveness of cognition and can lead to following poorly sourced recommended behavior [10]. 

Recent research has revealed that people whose primary source of COVID-19 information was social media experienced information overload (IO) [11], which subsequently impacted their information behaviors [12]. By definition, IO is “a physical and psychological distress that from human’s physical adaptive systems and decision-making process” (p. 326) [13]. It occurs when someone is unable to process all inputs from media engagements and then this causes ineffective learning or terminates his or her information processing [14]. In public health domains, health IO (HIO) refers to the situation where individuals fail to sensibly handle the amount of information relating to health issues during a given time frame [15]. HIO is more likely to occur when individuals encounter health information about an urgent or a public-concerned health issue, such as public health emergencies [16] and non-communicable diseases [17]. It is particularly true in the context of the COVID-19 infodemic. Individuals are inundated with a flood of information during daily media engagements; hence, the chance of suffering from IO is very likely increased, which can thus be recognized as a severe negative side effect. Therefore, it is essential to thoroughly understand the negative effects of social media use on individuals’ health information management during the infodemic period. 

## 2. Theoretical Foundation

IO has received significant scholarly attention because it is associated with health decision making and knowledge acquisition [10,15]. IO occurs when the quantity of information input is higher than the individuals’ ability to process information [18]. When the amount of information exceeds the available processing capacity, an individual has difficulty understanding it and may neglect a large amount of information that is vital for making health decisions [14,19]. Social media is an information avenue that enables individuals to engage with the latest information on many topics [20], but it is crucial to expand the body of knowledge on how information acquisition through social media can lead to IO in an infodemic.

The Cognitive Mediation Model (CMM) depicts how individuals are motivated to process information and gain knowledge from different media engagements. CMM posits that different motivations prompt individuals to pay attention to media and actively process and elaborate on the news and information they receive, which in turn influences the growth of knowledge and comprehension [21,22,23,24,25]. The core tenet of CMM shows that we should not consider knowledge gain to establish a direct exposure–effect process but, instead, a mediated approach. In the CMM framework, motivation is not directly associated with knowledge acquisition; instead, it subsequently fosters information engagement and cognitive elaboration [22]. For example, recent CMM studies in health domains found that different motivators triggered individuals to engage with H1N1 pandemic and cancer information; this, in turn, influenced their cognitive processing, knowledge gain, and behavioral intention [22,23,24]. In terms of applicability in the digital media environment, CMM has also been introduced on social media and the Internet to examine the public’s knowledge acquisition of scientific and health topics [25,26]. The underpinning role of the CMM framework in explaining the process of different knowledge acquisition in digital contexts is similar to its role in conventional media channels [21]. 

Although CMM has been used to study political and health information processing in various populations, e.g., [21,22,23,24,25], there are several gaps in the literature, particularly in understanding individuals’ information processing mechanisms during public health crises. First, CMM brings to light the functioning of elaboration in information processing, a positive cognitive approach to making sense of newly encountered information [27]. Effective elaboration helps individuals build connections among new information, existing knowledge, and previous experience. These connections influence their later knowledge development. Ideally, greater information engagement will increase the likelihood of positive elaboration, contributing to a higher level of knowledge [28]. In the context of COVID-19, however, finding specific answers is essential: Will more information engagements still drive a higher likelihood of positive cognitive elaboration? Will information engagements be effective when individuals encounter conflicting COVID-19 health information?

Second, since the beginning of the COVID-19 pandemic, Malaysia’s Ministry of Health has engaged with national media outlets and news agencies to swiftly convey targeted information regarding preventive measures to the public. However, this information obtained by the public has been heterogeneous. Aside from the credible information aired by official sources, the public also receives inauthentic information that triggers subreption and hatred, especially on social media. Diving into this complicated information environment, individuals’ cognitive processing ability may become interrupted or terminated, causing confusion, barriers, and uncertainties regarding the information [10]. Consequently, individuals become overwhelmed by too much information, limiting their ability to process it effectively [18]. Hence, it is reasonable to argue that when processing health information during an infodemic, the role of IO should not be neglected in CMM.

Third, a major similarity can be seen across several previous studies. Researchers assume that individuals already live in an environment featuring excess information on a specific topic [26,29,30]. Nonetheless, these studies fail to address the following: (1) how information processing causes IO, (2) the operationalization of IO in any information processing model, and (3) the predicted magnitude of IO due to information engagement.

Fourth, although previous studies have articulated the usefulness of CMM in explaining IO [8,24], they have failed to use the initial mechanisms of CMM, namely, the “motivation–attention–cognitive processing” approach for examining the consequence of information processing [18]. In other words, IO’s role in health information processing has yet to be identified. 

Taking these together, understanding IO based on the information processing paradigm, particularly CMM, is paramount for enhancing scholarly understanding of the negative consequences of information engagement on social media. In line with the prior CMM studies and the aforementioned questions, we aim to advance the CMM from three aspects. First, following prior CMM studies, we consider risk perception a key motivator for information engagement in the COVID-19 infodemic. Second, we argue that IO is associated with two types of information engagement on social media (receiving and expression). It clarifies the relationship in terms of how different information behaviors trigger IO. Third, echoing recent studies [31,32], we address one primary negative emotion during the COVID-19 pandemic: perceived stress, as a predictor of IO in health information processing. Thus, this current study extends CMM to health IO (HIO) by developing a conceptual HIO model in the context of the COVID-19 infodemic (as shown in Figure 1).

## 3. Literature Review and Hypotheses Development

### 3.1. Antecedent Motivator of Predicting Health Information Acquisition: Risk Perception

In public health emergencies, risk perception is one of the most important antecedents for predicting communicative and psychological actions [24,33]. However, the dimensions of risk perception during a pandemic are distinct from those in other health contexts, such as cancer and other health challenges [34,35]. Scholars have typically operationalized risk perception at the individual or personal level, emphasizing the cognitive and emotional dimensions [36,37,38]. Nevertheless, during the global COVID-19 pandemic, it has become inadequate to look only at how people conceive of a disease in terms of the risk to themselves (personal level risk perception). In fact, the entire population is at some degree of risk. Therefore, the risk perception regarding infectious health emergencies goes beyond personal to societal and global relevance [35,39,40]. Ample evidence of the relationship between different dimensions of risk perception and information engagement has been reported. For example, a comparative study found that Chinese college students perceived higher risk at the personal and societal levels than their American counterparts regarding the H1N1 pandemic, which was affected by their relevant media exposure and interpersonal discussion [39]. A recent study found that those who perceived higher risk at the individual and societal levels were more likely to seek information on the Zika virus, showing mobilized preventive intention; however, risk perception at the global level failed to demonstrate this relationship [35].

For the case of COVID-19, Dryhurst et al. [34] argued that researchers should consider risk perception not only in relation to well-established and model-driven constructs but also in terms of integrated logic from a range of schools, such as perceived likelihood, perceived seriousness, and temporal-spatial differences [41]. This approach extends the self–other risk difference dimension of Han et al. [39] and the three-level approach of Lee et al. [35] and provides a holistic way of measuring public risk perception during a pandemic. Previous studies on risk perception during the COVID-19 pandemic have paid little attention to different levels of risk perception (personal, societal, and global) or how these dimensions motivate people to engage on social media during a pandemic. However, this study follows and adopts a conceptualization of public risk perception during a pandemic involving perceived seriousness and likelihood at the personal, societal, and global levels [34]. 

### 3.2. Linking Risk Perception, Social Media Engagement, and Health Information Overload

The proposition of CMM indicates that health information engagement is significantly stimulated by health motivation [22,23]. Recent studies have found that risk perception is a strong antecedent motivator for health-related media attention [23,24]. For example, Lee et al. [23] showed that risk perception positively predicted media attention and interpersonal communication concerning breast cancer among Singaporean women. Likewise, Zhang and Yang [24] also reported that risk perception positively predicted information seeking and scanning behaviors regarding breast cancer information in China.

With the growth of the use of social media, particularly during the infodemic, studies have identified the development tendency of CMM—from examining attention on different media channels [42] to understanding other information behaviors (e.g., seeking and scanning) [24]. On the one hand, information seeking is increasingly acknowledged as a purposeful activity for obtaining information from certain sources and as an activity that requires effort to obtain information outside the typical exposure pattern [43]. Information scanning, on the other hand, refers to the amount of attention paid to the media and is usually categorized as a passive information acquisition behavior [44]. In particular, scholars have noticed that it can be challenging to conceptualize information scanning [45,46]. During daily digital media usage, individuals may encounter news notifications through social media or online news applications that may not be the information they are seeking. The decision whether to follow notification and read a full-length article requires active cognitive effort. Hence, information scanning in the digital environment cannot be completely understood as unintentional or passive information exposure. Additionally, it entails a decision-making process that actively seeks out the information [47]. Yoo et al. [48] simplified the information seeking and scanning mechanism by highlighting communicative actions rather than identifying actions as active or passive to avoid the definitional conflict. Yoo et al. [48] explained that when social media is widely accessed, particularly during a crisis, users are more likely to be exposed to messages that may influence their perceptions and preventive behaviors related to crisis events. However, users not only receive messages but also express their thoughts to their social circles. Consequently, during public health emergencies, the public may play a crucial role in message expression through social media, for instance, by creating content, amplifying and commenting on traditional news articles [49]. For the MERS pandemic, Yoo et al. [48] documented that individuals who expressed more relevant information on social media had better self-efficacy than those who expressed less. Those who received more MERS information can be expected to have greater perceived threats and stronger preventive intentions. Another study found that people who used Facebook to express more information concerning dust pollution had a higher level of preventive intention [15]. By contrast, people who received more relevant details concerning dust pollution showed a greater intention to comply with government recommendations. Therefore, following the conceptualization of Yoo et al. [20], linking risk perception and information engagement on social media, we postulate:

**Hypothesis 1** **(H1).***COVID-19 risk perception is positively associated with (a) receiving and (b) expressing COVID-19 information on social media*.

HIO was initially conceptualized in cancer risk communication [15]. There has been a noticeable shift in research focus, from creating instruments for a specific health concern to a thorough examination of the relationship between HIO and other health-related factors [26,50]. Scholars have been examining the effects of HIO since the commencement of earlier versions of the Health Information National Trend Survey (HINTS). A 2007 HINTS study found that sex, age, perceived health status, and socioeconomic status were all associated with CIO. That study also showed that people with no history of depression or only mild depression were more likely to feel overloaded [51]. Recently, a study found that individuals who experienced greater IO were reluctant to receive an annual medical check-up and were less knowledgeable about sun-safe behaviors [17].

In addition, several empirical studies have also been performed to understand the role of HIO in other health domains, including the COVID-19 pandemic. Notably, researchers highlighted the association between media engagement and perceived HIO. For instance, a cross-national survey study found that individuals receiving information from mass media were more likely to experience high COVID-19 IO than those who received information via social media [52]. Another study in Finland also articulated that receiving COVID-19 information on social media caused the public to feel overwhelmed [53]. Therefore, it is crucial to clarify the underlying mechanism of HIO during the pandemic, particularly cause–effect relationships related to information behaviors. Instead of studying HIO as an external condition, a different approach is taken here, operationally defining it as the consequence of an amount of information exceeding the human information processing capacity [18]. In line with this reasoning, we propose:

**Hypothesis 2** **(H2).***(a) Receiving and (b) expressing COVID-19 information on social media are positively associated with HIO*.

### 3.3. Linking Social Media Engagement, Negative Emotion, and Health Information Overload

In infectious disease outbreaks, individuals living within a media environment are exposed to emotional content [32]. Such content could produce different responses in different individuals. Researchers have found that health information, especially information from online channels, is largely emotionally framed [32,54]. The number of confirmed disease cases or reporting of mounting death tolls is highlighted, along with the presentation of manipulated information and the spreading of misinformation. This information triggers cognitive uncertainty about the disease and negative self-relevant emotions, including fear, stress, and worry [55,56]. For instance, during the MERS outbreak, fear and anger were dominant negative emotions among the general public [57]. Another study also found that fear and anger triggered perceptions of personal risk regarding MERS, which in turn influenced preventive behaviors [32]. Yang et al. [58] documented that exposure to virus-related information in media could cause fear among female Americans.

Moreover, self-relevant emotions have been highlighted in health communication studies in relation to cancer. Chae [55] found that fear and worry were positively associated with cancer information use. Fear was also positively related to cancer information avoidance, whereas worry is negatively associated with avoidance. Scholars have also observed the effects of negative emotions and intentions and the use of preventive measures in COVID-19. For example, social media usage in Chinese populations may lead to significant worry concerning the outbreak, where worry is positively associated with preventive behaviors [59]. In Thailand, a study indicated that the longer the time spent on information engagement on social media, the greater the chance one might suffer from anxiety and fear [60].

Although COVID-19 studies usually focus on negative emotions, such as worry, fear, and anger [29,30], Lwin et al. [61] found that the percentage of fear-centered Twitter posts drastically decreased from approximately 60% at the end of January 2020 to less than 30% in early April 2020. The trend in angry Twitter posts was relatively stable within those 3 months. Therefore, as the world has been suffering from this pandemic for more than a year, it seems clear that stress, a long-term negative emotion, can become more arresting than worry or fear. Empirical studies have identified different roles of perceived stress. For example, one study found that social media usage and interpersonal communication about COVID-19 issues were positively associated with stress [62]. Another study showed that fatalism regarding COVID-19 is a strong and positive predictor of stress [31]. Hence, we predict:

**Hypothesis 3** **(H3).***(a) Receiving and (b) expressing COVID-19 information on social media are positively associated with perceived stress*. 

Following Schmitt et al. [18], we consider IO a negative consequence of information engagements. We seek to understand the role of negative emotion in predicting IO. In the literature, the associations between negative emotions and information avoidance, a consequence of HIO, have been examined [50]. For instance, fear of cancer causes cancer information avoidance, but worry does not [63]. In the COVID-19 pandemic, Song et al. [57] reported that anxiety, a dominant negative emotion, is associated with information avoidance instead of sadness. Thus, we investigate the direct association between perceived stress and HIO. We postulate:

**Hypothesis 4** **(H4).***Perceived stress is positively associated with HIO*.

## 4. Materials and Methods

### 4.1. Sample and Data Collection

This study was approved by the Institutional Review Board of the authors’ affiliated institution (UPM/TNCPI/RMC/JKEUPM/1.4.18.2 (JKEUPM)). The questionnaires were prepared in English, Malay, and Mandarin. We first drafted an English version and subsequently conducted back translation to create the Malay and Mandarin versions. We included five variables measured in the hypothesized model and the demographic variables of gender, age, ethnicity, religion, and education level in the questionnaire. The study was conducted from May to June 2021, during Malaysia’s third round of MCO (i.e., full lockdown). To overcome the challenges posed to data collection during this period, the recruitment method adopted in this study mirrored the convenience sampling method used by Azlan et al. [64] in their study assessing knowledge, attitudes, and practices in the Malaysian public regarding COVID-19. Weblinks to the online questionnaires using Google Forms were disseminated through social networking sites to recruit respondents. The snowballing method was also adopted to increase the sample size. The final sample size was 676 (*n* = 676), with no missing data. This study used an a priori sample size calculator for structural equation modeling (SEM). Given the number of observed (N = 16) and latent (N = 3) variables, the anticipated effect size (*d* = 0.30), the desired probability (*p* = 0.05), and intended statistical power (0.80), a minimum sample size of 123 was required. Our sample of 676 surpassed the recommended minimum sample size for sampling adequacy.

In our sample, most respondents were women (56.2%), and the average age was 32.87 (SD = 10.60, ranging from 18 to 68 years). The sample consisted of 48.5% Malays and 39.8% Chinese. By religion, 49.4% were Muslims, 21.9% were Buddhists, and 20.0% were Christians. Most respondents had attended college (diploma and above, 86.2%).

### 4.2. Measurements

To assess COVID-19 risk perception, respondents were asked to respond to seven items using a 6-point scale ranging from 1 (strongly disagree) to 6 (strongly agree). The conceptualization of Dryhurst et al. [34] was adopted to measure this construct by modifying items from studies that examined risk perception in previous pandemics [29,33]. Example items included “The problem of the COVID-19 outbreak is important to me” and “I am worried that I will be infected with COVID-19 in the future”. We averaged the responses to create one scale. A higher score indicates a higher risk perception (*α* = 0.93, *M* = 5.12, *SD* = 0.89).

To assess information engagement regarding COVID-19 on social media, respondents were required to respond to two single items derived from previous research [48]: “How often did you receive/express COVID-19 information on social media platforms (e.g., Facebook, Twitter, Instagram, WhatsApp, Telegram, WeChat…) during the last 7 days?” The responses were measured on a 6-point scale ranging from 1 (not at all) to 6 (very much). Higher scores indicated higher social media engagement (*M_receving_* = 5.54, *SD_receiving_* = 0.66; *M_expressing_* = 5.21, *SD_expressing_* = 0.84).

Perceived stress was measured with two items on a 6-point scale ranging from 1 (strongly disagree) to 6 (strongly agree) [65]. These were (1) “Currently, I feel so down in the dumps that nothing could cheer me up” and (2) “Currently, I feel downhearted and blue”. Higher scores indicated higher stress levels (*α* = 0.84, *M* = 4.84, *SD* = 1.07).

Guided by the measurement used by Jensen et al. [15], HIO was assessed using a modified instrument from Costa et al. [66], altered by replacing the word “cancer” with “COVID-19”. This instrument featured five items with responses on a 6-point scale ranging from 1 (strongly disagree) to 6 (strongly agree), such as “There are so many different recommendations about preventing COVID-19, so it’s hard to know which ones I should follow” and “Information about COVID-19 all starts to sound the same after a while”. Higher scores indicate higher levels of IO (*α* = 0.91, *M* = 5.44, *SD* = 0.66).

### 4.3. Data Analysis

Bivariate correlations between measured variables are shown in Table 1, and confirmatory factor analysis was conducted with the results shown in Table 2. SEM was performed with the lavaan package in R [67]. We used maximum likelihood estimation to examine the pathway coefficients of the hypothesized model. To establish the proposed model and evaluate its fit, the following criteria were considered: (1) relative chi-square ($x^{2}$/df), (2) comparative fit index (CFI), (3) Tucker–Lewis index (TLI), (4) root mean square error of approximation (RMSEA), and (5) standardized root mean square residual (SRMR). If the model has a good statistical fit with the data, the value of the relative chi-square should fall between 1.0 and 5.0 [68], and the CFI and TLI values need to be higher than 0.95 [69], RMSEA should be close to 0.06, and SRMR values should be less than 0.08 [70].

## 5. Results

From the SEM results, our hypothesized model showed an acceptable model fit, $x^{2}$/(99) = 3.96 (*p* < 0.001), CFI = 0.97, TLI = 0.96, RMSEA = 0.06 (*p* < 0.001, 95% CI: 0.059–0.073), SRMR = 0.03. This model explained 81.0% of the variance in information receiving ($R^{2}\;$= 0.81), 70.0% in information expressing ($R^{2}$ = 0.70), 38% in perceived stress ($R^{2}$ = 0.38), and 73.0% in the outcome variable, HIO ($R^{2}$ = 0.73).

H1 proposed that COVID-19 risk perception was positively associated with receiving (H1a) and expressing (H1b) COVID-19 information on social media. As shown in Figure 2, COVID-19 risk perception had a significant positive relationship with information receiving (β = 0.90, *p* < 0.001) and information expressing (β = 0.84, *p* < 0.001), indicating that the higher the risk perceived regarding the pandemic, the higher the likelihood of receiving and expressing COVID-19-related topics on social media. Thus, H1a and H1b were supported.

H2 postulated that information receiving (H2a) and information expressing (H2b) on social media were positively associated with HIO. The result (shown in Figure 2) demonstrated a significant positive relationship between information receiving and HIO (β = 0.84, *p* < 0.001), which means that when someone receives additional information on social media about COVID-19 issues, their level of IO increases, supporting H2a. However, the results failed to support H2b, as there was no significant association between information expression on social media and HIO (β = 0.05, *p* = 0.23), meaning expressing more COVID-19 information on social media would not directly trigger HIO.

H3 hypothesized that information receiving (H3a) and information expressing (H3b) were positively associated with perceived stress. The results supported the hypothesis, as both information receiving (β = 0.44, *p* < 0.001) and information expressing (β = 0.21, *p* < 0.001) on social media were positively associated with perceived stress, indicating that when individuals received and expressed more COVID-19 information on social media, their stress level increased. Thus, H3a and H3b were supported.

H4 posited that perceived stress was positively associated with HIO. The result showed a significant positive relationship between perceived stress and HIO (β = 0.54, *p* < 0.001), demonstrating that the greater the stress level, the greater the IO among the public, meaning that H4 was supported.

## 6. Discussion

This study investigated the underlying pathways causing a negative information processing consequence, HIO, during the COVID-19 pandemic based on a derivate CMM among Malaysian survey respondents. Results showed that HIO among our respondents in the context of COVID-19 could be explained using a stepwise and mediated information processing model. In the model, COVID-19 risk perception acted as an essential motivator. It was associated with information engagements on social media platforms, which were linked to negative emotion (i.e., stress) and the final informational outcome, HIO.

First and foremost, our online survey results involving Malaysian respondents resonated with the postulation regarding the role of risk perception in well-developed health information management frameworks. It serves as a determinant of information behaviors and problem-solving mechanisms across different public health concerns. In our case, risk perception mobilized Malaysian respondents to engage with COVID-19 health information from various social media platforms, such as Facebook, Telegram, and WeChat. We proposed and included one type of context-specific risk evaluation termed COVID-19 risk perception. It acted as the antecedent motivator in the hypothesized model. This variable contains individuals’ risk beliefs, called perceived risk estimation, from three dimensions, which are personal, societal, and global levels. COVID-19 risk perception thoroughly captured individuals’ subjective estimation about the likelihood of contracting COVID-19 infection not only for themselves but also for others in their communities and avenues far away. This conceptualization was aligned with the nature of an infectious disease outbreak [35]. Unlike other non-communicable diseases, the likelihood of COVID-19 infection is no longer a personal threat but a societal or global issue. The degree to which an individual perceives others around or far away from them are in danger of the infection would very likely impact their likelihood of taking preventive measures [35,39]. 

In terms of health information management, our results indicated that COVID-19 risk perception positively predicted receiving and expressing COVID-19 information on social media channels among Malaysian respondents. It means that the higher the degree that respondents estimate their own perceived likelihood of contracting the virus along with the likelihood for others, the higher the frequency they would receive and discuss COVID-19 health information on social media platforms. Of note, this finding was consistent with previous studies on other public health concerns, such as breast cancer knowledge acquisition among females from China and Singapore [23,24], and echoed with several health information processing frameworks [22,71]. For instance, apart from CMM, the Risk Information Seeking and Processing Model (RISP) [72] and the Planned Risk Information Seeking Model (PRISM) [71] also included risk perception as a vital concept to explain how information behaviors are formed through a psychological angle. Perceived risk indirectly triggers information seeking intention, behaviors, and systematic information processing through increased affective responses to the risk. Therefore, the role of risk perception is crucial for helping individuals equip themselves with potentially useful information about the perceived threat, which would likely influence the subsequent coping strategies and preventive adoptions. 

Furthermore, our results showed that the receiving and expressing of COVID-19 health information on social media differ concerning the prediction of HIO. A direct and significant positive association was seen between information reception and HIO. However, the model failed to report a significant relationship between information expression and HIO. In response, when someone receives more COVID-19 information from social media, they become more likely to feel overloaded by the information; nevertheless, expressing or sharing it through interpersonal networks on social media does not directly result in becoming overwhelmed. This is a logical finding, especially given the support from the initial theoretical mechanism in CMM. Attending to messages from mass media could foster individuals’ cognitive processing of the received information [21]. Notably, the proposition in the original CMM considers cognitive processing as “elaboration” or termed “elaborative processing”, which refers to a positive way how individuals deal with newly encountered information. Elaboration links the newly received information with someone’s previous personal experiences and existing knowledge, producing a new understanding of a specific topic [22]. However, in the context of our study, as individuals were situated in an information environment containing ambiguous information and misinformation about COVID-19, they were more likely to experience negative feelings or psychological reactions when they received a flood of information on social media every day, instead of spending cognitive efforts to digest the information systematically. We can say that individuals’ usual elaborative processing ability is very likely disrupted and interrupted during the infodemic. Subject to this phenomenon, if individuals receive too much COVID-19 health information on social media, HIO is reasonably triggered. Therefore, although engaging with media is usually beneficial for gaining health knowledge, it may become disruptive if too much information is available, causing people to be overwhelmed as the information is assessed [30,56]. Our results echoed Jiang and Beaudoin’s [26] finding regarding the pathway from Internet-based health information acquisition to the level of health literacy among Americans. Individuals tend to hold a lower level of health literacy if they are overwhelmed with the amount of health information obtained from the Internet [26]. This result also aligned with a previous study’s definition and conceptualization of HIO [10]. HIO results from failing to process health information that is input, causing confusion and ineffective learning on specific topics. It highlights the negative effects of input information or exposure to the media [15]. 

Furthermore, we found that among Malaysian respondents in our sample, their posting, sharing, or discussing COVID-19 health information on social media did not directly cause HIO. A plausible explanation is by looking at the nature of information expressing [20]. This information behavior can be explained through the expression–effects paradigm [73]. Individuals should organize the messages in their minds before expressing thoughts, ideas, or opinions regarding specific topics. Indeed, organizing what messages should be delivered and how to send these messages to receivers involve active information processing [48]. Active information processing can also be observed along with discussing COVID-19 topics with interpersonal networks on social media. This process involves real-time interactions, which can reduce the level of existing anxiety and uncertainty about the COVID-19 outbreak [74]. Thus, it makes sense that information expressing did not directly predict HIO.

Another important finding was that receiving and expressing COVID-19 topics on social media were both positively associated with perceived stress. In other words, the more individuals received and expressed COVID-19 information on social media, the higher the level of stress induced. During the COVID-19 infodemic period, the flood of conflicting information directly interrupted individuals’ cognitive processing of the information. It caused long-term mental health issues, such as long-term stress, in the case taken in this study. Regardless of the mechanisms of the specific information behavior (receive or express information), long-term stress was likely to be perceived through both pathways. As COVID-19 became a years-lasting global pandemic, individuals started to suffer from continuing psychological issues, such as anxiety, frustration, and especially long-term stress [61,75]. In Malaysia, during data collection, the public was situated in a physical lockdown and a complex information environment for more than a year, with COVID-19 information behaviors dominating everyone’s daily lives, thus causing them to experience long-term stress [31,76]. Several recent studies also discovered the roles that stress played during this pandemic. For instance, when perceiving a higher stress level among college students in Turkey, they were more likely to think that their current life was meaningless and had a higher chance of experiencing psychiatric symptoms [77]. Furthermore, another study reported that a higher level of COVID-19 stress decreases mental well-being among Palestinian adults [78]. 

The results revealed that individuals perceive stress from their information behaviors on social media, filling in gaps left by previous studies on the conceptualization of perceived stress. Similarly, the link between information acquisition and other negative emotions was also examined in recent studies. For example, engaging in risk information on social media caused individuals to feel more anger and fear during the MERS outbreak [32]. Yang et al. [58] also found that media exposure was a key determinant of fear during the Zika outbreak in the US. Hence, our results highlighted the significance of negative emotions in health information processing, specifically perceived stress. Meanwhile, the finding is partially consistent with the framework of Ngien and Jiang [31]: social media usage during the COVID-19 pandemic causes stress indirectly. Our model finally showed a positive association between perceived stress and HIO, indicating that the greater the perceived stress from health information engagement, the higher the likelihood of experiencing HIO during the COVID-19 infodemic. This finding showed that the effects of negative emotions are not only limited to psychological symptoms but also information processing, bridging information behaviors, and their consequences [57,63]. Furthermore, although our model did not detect a significant direct relationship between information expressing and HIO, expressing COVID-19 health information on social media triggered perceived long-term stress (as discussed above), and perceived stress was then, in turn, positively associated with HIO. Therefore, we can conclude that the nature of the infodemic is more potent than the commonly recognized mechanism of specific information acquisition approach in terms of impacting individuals’ health information management strategies and mental health conditions.

## 7. Implications, Limitations, and Suggestions

This study provides several theoretical implications. First, to our knowledge, it is the first study to involve HIO as an outcome variable in the CMM and validate its cause–effect relationships. Grounded in CMM, this study showed that other than knowledge gain, HIO, as one type of negative information processing consequence, can also be explained by the CMM paradigm. We sought to fill the theoretical gaps left by Chae et al. [10], who correlated several factors from different dimensions with CIO. In this study, we conceptualized that the cause of feeling overwhelmed by health information is a failure of effective information processing. Thus, the cause–effect relationships of HIO are explained in an information processing model, which provides complements and new dimensions for the HIO literature. Future research can apply our model in diverse health contexts. Second, we extended the conceptualization of risk perception as a motivator for information engagement. We considered it reasonable to follow Lee et al. [23] and Zhang and Yang [24] because, during the global COVID-19 pandemic, differences in risk perceptions could lead to different cognitive responses. Our study shows that different dimensions of risk affected information engagements on social media. Thus, future research about the motivation factors for acquiring information on social media can follow our categorization of risk perception. However, justifications regarding different health issues need to be highlighted. Third, we argue that different information behaviors might have different effects when using social media in health contexts. To avoid the definitional conflict in information seeking and scanning, we adopted Yoo et al. [48]’s conceptualization of information receiving and expressing. Thus, this result extends the categorization of information acquisition on social media and its potential effect on human information processing capacity. 

This study also contributes to public health practices, especially health information management. By analyzing COVID-19-related HIO through a nationwide survey in Malaysia, we may draw the attention of health authorities to the significance and possible consequences of HIO. This may help practitioners develop a comprehensive picture of how the public is affected by HIO. As a result, intervention strategies can be designed and implemented to deal with HIO. For example, the authorities should monitor online information and ensure its quality and accuracy. Health educators should engage the public on social media by sharing authorized and up-to-date COVID-19 information, correcting misinformation and offering mental support. Implementing this line may reduce the feeling of being overwhelmed and tired of COVID-19 health information among the public.

This study had some limitations. First, we could only use a cross-sectional online survey with convenience sampling to recruit respondents as it was conducted during a lockdown. It failed to properly reflect the demographic structure in Malaysia, especially in terms of ethnicity. These factors limit the generalizability of our findings. Future studies may ideally use systematic and purposive samples to cover a broader diversity of respondents. Second, the present study obtained only a snapshot of the impact of HIO during the COVID-19 pandemic, failing to examine the trend and changes in HIO that may alter or differ from one period to another. Therefore, future studies should adopt a longitudinal method to evaluate the trends and changes in HIO. Third, the scope of this study is limited to examining social media in general rather than examining the effects of technological affordances (e.g., algorithms and interactivity) on individuals’ cognitive processing. It is worth noting that a wide variety of technological affordances provide different functions concurrently available on social media. Hence, there is a need for future research to investigate how CMM functions in terms of other types of media and a broader set of technological affordances to determine how they can lead to elaboration and learning. Forth, the conceptualization of information engagement may be biased as we only considered information engagement on social media, which was chosen because social media has been the main avenue through which the public has consumed information about the pandemic. However, the role of mainstream media cannot be ignored. Studies showed that the mainstream media is still the most important source of information for members of the public seeking COVID-19 updates in 2020 [79,80]. The original definition of HIO emphasizes the word “input information” [15], neglecting the media channels used. Hence, future research should examine information engagements on both traditional and digital media channels when considering HIO. Fourth, we only included perceived stress in the model from among negative emotions. Although stress is the primary long-term feeling in the COVID-19 context, future research should extend this model by including other emotions, such as trait anxiety [10], worry, and anger, to discover the differences in predicting HIO.

## 8. Conclusions

Of note, research on HIO is still in its early stages, particularly in the context of an infectious disease outbreak. To date, limited studies have concentrated on the underpinning mechanisms of how HIO is caused, primarily through a thorough information processing angle. This study thus broke new ground by postulating a conceptual model on HIO and empirically tested it by involving Malaysian respondents during the middle of the COVID-19 infodemic. Guided by the theoretical foundation from a well-established information processing framework, CMM, we considered risk estimation of the COVID-19 outbreak from three distinct dimensions (i.e., personal, societal, and global levels) as the motivational antecedent that fostered individuals to acquire COVID-19 health information on social media. Furthermore, we included two distinct information acquisition behaviors in our conceptual model—information receiving and information expressing. This postulation differs from the traditional approach of how researchers categorize health information behaviors based on the degree of cognitive efforts spent (i.e., information seeking or information scanning). According to the reception–effects paradigm and expression–effects paradigm, we solely concentrate on the actions that individuals performed to acquire health information, which simplified the organism of health information management. Interestingly, our results showed that receiving health information on social media directly caused individuals to suffer from HIO but not expressing. We also included a severe mental health condition and perceived long-term stress in the model. Results depicted that both information behaviors were positively related to perceived stress among our respondents, which subsequently triggered HIO. This phenomenon explained how the COVID-19 infodemic significantly impacts individuals’ psychological conditions and information management approaches. In sum, the conceptual model in this study explained a relatively comprehensive picture of how information behaviors can cause HIO during the COVID-19 infodemic through a stepwise and mediated angle. Therefore, we encourage future studies to involve other solid theoretical foundations, further expand this model in different public health settings, and improve the conceptualization and operationalization of HIO in health information processing.

## Figures and Tables

**Figure 1 ijerph-19-07991-f001:**
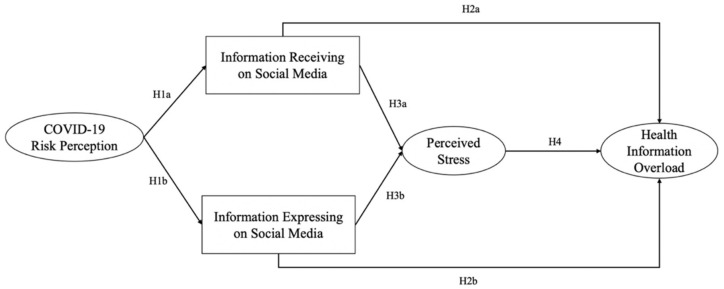
Conceptual model.

**Figure 2 ijerph-19-07991-f002:**
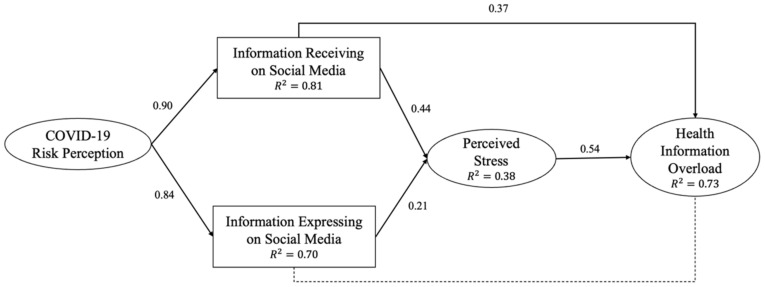
Conceptual model after analysis.

**Table 1 ijerph-19-07991-t001:** Bivariate correlations between measured variables (N = 676).

Variable Name	(1)	(2)	(3)	(4)	(5)
**(1)** COVID-19 Risk Perception	/				
**(2)** Information Receiving	0.87 **	/			
**(3)** Information Expressing	0.82 **	0.75 **	/		
**(4)** COVID-19 Stress	0.55 **	0.55 **	0.50 **	/	
**(5)** Health Information Overload	0.66 **	0.70 **	0.59 **	0.70 **	/

*Notes*: ** = *p* < 0.01.

**Table 2 ijerph-19-07991-t002:** Confirmatory factor analysis results of measured variables (N = 676).

Item Name	Factor Loading	*M*	*SD*
*COVID-19 Risk Perception* (α = 0.93)		5.12	0.89
RP1	0.86		
RP2	0.88		
RP3	0.83		
RP4	0.84		
RP5	0.82		
RP6	0.80		
RP7	0.66		
Information Receiving	/	5.54	0.66
Information Expressing	/	5.21	0.84
*COVID-19 Stress* (α = 0.84)		4.84	1.07
Stress 1	0.84		
Stress 2	0.87		
*Health Information Overload* (α = 0.91)		5.44	0.66
HIO1	0.82		
HIO2	0.84		
HIO3	0.80		
HIO4	0.84		
HIO5	0.83		

## Data Availability

Data sharing is available upon request.

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
