# Peer review of "The Effects of Receiving and Expressing Health Information on Social Media during the COVID-19 Infodemic: An Online Survey among Malaysians"

_ijerph, 2022, doi:10.3390/ijerph19137991_

Round 1
Reviewer 1 Report
I think the introduction should collect more information and present the study a little better.
I think more detail should be given on the following statement at the beginning of the methodology:
This study was approved by the Institutional Review Board of the authors’ affiliated 273
institution (UPM/TNCPI/RMC/JKEUPM/1.4.18.2 [JKEUPM]).
Having read the methodology, I think that in the introduction (including the title and abstract) the specific case study that is being carried out should be detailed. Or justify the reason for that sample.
I think that the conclusions should better connect everything addressed throughout the study.
Author Response
-
Thank you so much for this concern. We have improved the introduction by including the more detailed background information. First, we explained the impact of COVID-19 on people’s daily lives and how people stay connected and informed about the pandemic. Second, we highlighted that despite the widely acknowledged popularity and contributions of social media, the development of social media can create new problems, such as mis/dis/mal- COVID-19 information is freely spreading on various social media platforms without any censorship, including various conspiracy theories about the virus and vaccination, misleading information about how to treat COVID-19 infections and scientifically incorrect messages on preventive measures, and the effects of exposing to such information on people’s cognitive processing capacity—information overload. Kindly refer to the revised manuscript for the update. This part has been highlighted in yellow.
-
Thank you for this concern. First, we have amended the title to a more specific manner. We highlighted the study population in the revised title. Please refer to the revised manuscript for the changes. Regarding the second concern raised, we have mentioned in the abstract that this research is conducted among Malaysian respondents through an online survey.
-
Thanks for this suggestion. We have amended the original conclusion by elaborating more details we have done in this study. Please refer to the revised manuscript for updates.

Reviewer 2 Report
The topic is very interesting and actual. However, I emphasize two aspects that could be improved. On the one hand, the introduction of definitions of the main keywords, such as health information overload, information overload, etc. And, on the other hand, the main media channels.
Author Response
- Thank you very much for the suggestions. We have improved the introduction part and re-wrote some parts for the introduction in the revised manuscript. Please see the attachment.

Reviewer 3 Report
please see the attachment

Author Response
-
Thank you so much for the suggestion. We have improved the introduction section by incorporating more detailed background information and some explanations for the terms used in this study. Please refer to the revised manuscript for the changes. It has been highlighted in yellow.
-
Thank you again for the concern. We have improved the write-up for the discussion section. We elaborated more detailed explanations to discuss the results our model obtained. More comparisons and examples have been added in the revised manuscript as well. Please refer to the part highlighted in yellow for the update.

Reviewer 4 Report
I want to thank the authors for the opportunity to read their work. The article “From a pandemic to an Infodemic: the effects of receiving and expressing covid-19 information on Social Media”. The paper represents an exciting contribution to various disciplines, from health information, psychology, and medicine to communication studies and social sciences, among others. For that reason, the following comments are driven to increase the overall quality, clarifying some points and offering an external view on it.
1. Title: in its current form, it infers a generalization and does not fit with the article's scope and limitations. Adjusts must be done to reflect the study carried out.
2. Theoretical foundation: there is a first mention of CMM (line 54) and does not explain that it makes reference to the Cognitive Mediation Model. More important than it: there is no reference to other studies on CMM and Social Media. This must be solved, by incorporating literature/references from other studies on social media and CMM
3. Social Media: in general, this is an aspect that should be improved in the theoretical foundation and the literature review. We suggest that the authors assume what you consider social media is and present some data (from official reports) on social media penetration in Malaysia. This is also crucial to figure out (or to underline) sampling.
4. Social Media literature: in line with the previous comments, anthropologists such as Daniel Miller (“Why we post”), sociologists (e.g danna boyd, Cristian Fucks or Manuel Castells ) or media scholars (e.g. P. Napoli, in “Social Media and The public interest) have discussed several aspects on why people post, the limits of web 2.0 or the public interest on it. In this sense, we recommend to the authors define better what is your approach to social media and the idea of users as authors (there are different theories, from prosumers to self-mass communication, among others). The mentioned authors are not a recommendation to be followed. My strong suggestion here is to clarify what you consider SM and these related behaviors, etc.
5. Algorithms are a substantial part of visible content in social media and the user’s production activity. Hence, we recommend you discuss it (although briefly) and check if it would impact your hypothesis. If not, an argumentation on why it would be necessary.
5. IO: to which extent can we relate (or not) it with misinformation and the so-called information disorder? Is there any link with the “infodemic”? Clarifying it will help readers to navigate through your arguments.
6. Avoid generalizations such as line 163 (“with the exponential growth of the use of social media”) without references or data.
7. Also along the discussion, avoid generalizations to the social media in general (the sampling is geographically limited).
8. Conclusions are limited. We suggest expanding it, explain the final conclusions of your study (what we get with it).
Author Response
- Thank you for this concern, we have amended the title to a more specific form in terms of the research context and target respondent.
-
Thank you for this comment, and we are sorry for this mistake. We have put the full name for CMM, the Cognitive Mediation Model, in the first place it appeared, and a brief introduction for CMM is provided in the revised manuscript. We also added a few CMM references relating to health issues and particularly in the context of social media. Please refer to the revised manuscript for the changes; it has been highlighted in yellow.
-
Thank you so much for this concern. Initially, we conceptualized this study based on the declaration of infodemic by the World Health Organization (WHO) and studies published during the early stages of the outbreak. Previous researchers highlighted that those popular social media platforms have become the mainstream channels where mis/disinformation has been spreading since early 2020. Subject to the advanced digital technologies, the Internet, along with several social media platforms, have become the predominant information channels for the mass public to access health information and obtain health knowledge. In Malaysia, the location of this study, we have referenced some recent statistics sources from the Malaysian government, showing that until 2021 (the year when the COVID-19 outbreak was extremely severe in this country), there are 86% of the total population are active daily social media users (“Malaysia has 28 million social media users”, 2021). Commonly used social media platforms are WhatsApp, Facebook, Twitter, Telegram, and WeChat. We have included these widely used social media platforms in our survey instrument. Of note, one market data service company, Statista, reported that Malaysia ranked 5th in social network penetration rate worldwide in 2021. Social media platforms eventually serve as one unreplaceable role in Malaysians’ daily life, especially during the long period of lockdown, physical isolation, and work from home (WFH) period in the year we were surveying. Health communication researchers commonly recognized this phenomenon and conducted highly relevant studies involving samples from other parts of the world. Therefore, we particularly embarked on the information behaviors on social media platforms among Malaysians in terms of how these behaviors would cause negative consequences. We list a few relevant articles below for your further reference.
-
- Viswanath, K., Lee, E. W., & Pinnamaneni, R. (2020). We need the lens of equity in COVID-19 communication. Health Communication, 35(14), 1743-1746. https://doi.org/10.1080/10410236.2020.1837445
- Su, Y., Borah, P., & Xiao, X. (2022). Understanding the “infodemic”: social media news use, homogeneous online discussion, self-perceived media literacy and misperceptions about COVID-19. Online Information Review, online first publication. https://doi.org/10.1108/OIR-06-2021-0305
- Ngien, A., & Jiang, S. (2021). The effect of social media on stress among young adults during COVID-19 pandemic: Taking into account fatalism and social media exhaustion. Health Communication, online first publication. https://doi.org/10.1080/10410236.2021.1888438
- The Star (2021). Malaysia has 28 million social media users as of January 2021, says Comms Ministry sec-gen.https://www.thestar.com.my/news/nation/2021/09/22/malaysia-has-28-million-social-media-users-as-of-january-2021-says-comms-ministry-sec-gen
-
-
Thank you very much for this concern. As social media platforms became the primary information sources for individuals during the pandemic outbreak, we consider mobile applications that contain functions such as engaging with user-generated content, posting, commenting, liking, and forwarding the content, as well as instant messaging among social networks. Thus, we selected commonly used social media platforms among Malaysians in this study, including Facebook, WhatsApp, Twitter, Telegram, and WeChat. We emphasized the actions that individuals usually do on social media, receiving and expressing information, instead of following the traditional approach to differentiate whether and how much cognitive effort they spent to acquire relevant information. Our conceptual approach simplified the organism about information behavior on social media.
-
As the scope of this study is limited to examining social media in general, rather than examining the effects of technological affordances (e.g., algorithms and interactivity) on individuals' cognitive processing; hence, we have highlighted the limitations and future recommendations in the manuscript.
-
In this study, we understand that information overload is a severe consequence of information acquisition in the context of the COVID-19 infodemic. During the infodemic, numerous forms of mis/disinformation were freely spreading on social media platforms, and this existing and actual situation can cause adverse outcomes to the public, especially information overload, a direct informational outcome. Therefore, we took infodemic as a general background to analyze the underlying mechanisms in terms of a comprehensive CMM-guided pathway from motivation for COVID-19 information engagement to the outcome, information overload.
- We have removed the overgeneralized word “exponential” in original line 163 in the revised manuscript.
-
Thank you for this suggestion. We have added relevant words to specify the scope of our study and avoid the overgeneralization of social media in general in the part of the discussion. Please refer to the revised manuscript for updates.
-
Thank you so much for this suggestion. We have expanded the length of the conclusion. Please refer to the revised manuscript for the update.

Round 2
Reviewer 4 Report
Thank you for the authors' kindly response and the revisions that made the paper more sharpness for an interdisciplinary audience and more accurate in general.